# Exploring the prevalence of severe obesity in a large teaching hospital

Ephrahim E. Jerry[1]ↄ*, Miriam A. Scheurwater[2]ↄ, Janine J.P. Ghielen[2],
Ilse van Donkelaar[2], Dennis van Veghel[2], Lukas R.C. Dekker[3], Simon W. Nienhuijs[1]

**1** Department of Surgery, Catharina Hospital Eindhoven, Eindhoven, EJ, The Netherlands, **2** Value based Health Care, Catharina Hospital Eindhoven, Eindhoven, EJ, The Netherlands, **3** Department of Electrical Engineering, Eindhoven University of Technology, Eindhoven, AP, The Netherlands

ↄ Authors contributed equally to this work.
* ephrahim.jerry@catharinaziekenhuis.nl

## Abstract

### Background

Obesity remains a critical public health issue in the Netherlands, where 50.2% of the population has a body mass index (BMI) over 25 kg/m², 15.6% exceeds 30 kg/m², and 3.5% meets the criteria for severe obesity (BMI ≥ 35 kg/m²). This study aimed to describe the prevalence and distribution of severe obesity across medical specialties in a large Dutch teaching hospital, to assess whether certain departments carry a disproportionate burden compared to national averages.

### Methods

A retrospective cohort study was conducted at Catharina Hospital in Eindhoven, using patient data from 2022 and 2023. Patients with a history or current treatment for Metabolic Bariatric Surgery (MBS) were excluded. The highest registered BMI during the study period was used for analysis. Data was analyzed across specialties with attention to procedural volume and hospital burden. Additionally, all surgical procedures were reviewed in patients with BMI ≥ 35 kg/m².

### Results

In total, in 8.0% of all hospital visits patients had a BMI ≥ 35 kg/m², compared to 3.5% in the general population. Specialties such as pulmonology (12.8%), internal medicine (10.4%), and orthopedic surgery (9.8%) showed a notably higher prevalence. Furthermore, a total of 5,472 surgical procedures were performed on patients with severe obesity, representing 13.5% of all surgical procedures. Common procedures included percutaneous coronary intervention in cardiology, laparoscopic cholecystectomy in general surgery, coronary artery bypass graft in cardiothoracic surgery, and total knee arthroplasty in orthopedic surgery.

**Data availability statement:** The data underlying the results presented in the study are available from the corresponding author, they also have been uploaded as Supporting information.

**Funding:** The author(s) received no specific funding for this work.

**Competing interests:** The authors have declared that no competing interests exist.

## Conclusion

Approximately one in thirteen hospital patients presents with severe obesity, and undergoes over 2,700 procedures annually, reflecting the scale at which severe obesity intersects with hospital-based healthcare. These findings emphasize the need for heightened awareness of obesity-related healthcare demands and the need for further research into the impact of severe obesity on surgical outcomes and use of hospital resources.

## Introduction

Obesity is a global public health crisis with profound implications for healthcare systems. Traditionally defined as excessive fat accumulation that poses a health risk, it arises from a complex interaction of genetic, behavioral, and environmental factors. Recently, the Lancet Commission proposed a refined definition of clinical obesity as 'excess or dysfunctional adiposity that impairs health', independent of body mass index (BMI) [1]. This evolving definition underscores the complexity of obesity as a disease entity and may influence how its prevalence is measured and addressed in clinical settings. According to the World Health Organization (WHO), over 2.5 billion adults have overweight, and nearly 900 million are living with obesity [2]. In the Netherlands, nearly half of all adults have a body mass index (BMI) above $25\,kg/m^2$, and an estimated 3.5% meet the criteria for severe obesity (BMI ≥ $35\,kg/m^2$) [3,4].

Although the association between obesity and chronic diseases such as type 2 diabetes, cardiovascular disease, obstructive sleep apnea, and osteoarthritis is well-established, its distribution and impact within hospital populations remain insufficiently characterized [5]. Patients with severe obesity are frequently overrepresented in clinical specialties where excess weight exacerbates disease severity and complicates treatment. These include surgery, orthopedics, cardiology, and pulmonary medicine, where obesity is associated with impaired wound healing, increased anesthetic and perioperative risk, reduced joint and pulmonary function, higher infection rates, challenges in imaging and medication dosing, and greater logistical complexity [5,6].

Obesity plays a central role in both the development and progression of numerous chronic diseases, either directly—such as through biomechanical stress in osteoarthritis—or indirectly, via metabolic dysregulation leading to conditions like type 2 diabetes and cardiovascular disease. This results in a higher prevalence of obesity in hospital settings, particularly in specialties treating obesity-related conditions. Although obesity is also known to increase the risk of treatment complications, such as surgical site infections and delayed recovery, this study primarily focuses on the distribution of severe obesity across hospital departments and its implications for healthcare demand, rather than on clinical outcomes or postoperative complications [7,8]. Obesity also complicates abdominal wall hernia repair, due to elevated intra-abdominal pressure and impaired fascial healing. Previous research showed that patients with a BMI ≥ $35\,kg/m^2$ had nearly three times higher risk of recurrence after surgery [9].

Despite these clinical associations, there is limited data on how severe obesity is distributed across hospital departments, and whether certain specialties carry a disproportionate burden compared to general population trends. Importantly, obesity is increasingly recognized as a driver of elevated healthcare demand and procedural complexity, posing significant challenges for hospital capacity planning and multidisciplinary care coordination, and placing additional strain on healthcare resources [2,10].

This research aims to describe the prevalence and distribution of severe obesity across hospital specialties and to assess the representation compared to national population trends.

## Methods

### Study design and setting

This retrospective cohort study was conducted at Catharina Hospital, a tertiary referral center in Eindhoven, the Netherlands. In the study electronic health records from 2022 and 2023 were analyzed to determine the prevalence and distribution of severe obesity (defined as BMI ≥ 35 kg/m$^2$) across hospital specialties. The prevalence was compared to the prevalence of severe obesity in the general Dutch adult population, according to national statistics (3.5%) [3].

### Study population

The study included adult patients (≥18 years) with an active Diagnosis Treatment Combination (DBC) registration — the system used in the Netherlands to classify and reimburse healthcare based on a patient's diagnosis and the associated treatment trajectory. The DBC system is comparable to diagnosis-related groups (DRGs) used in other countries and enables structured data analysis across medical specialties and conditions [11], based on a patient's diagnosis and the corresponding treatments provided, and a recorded BMI. Patient counts per specialty reflect unique individuals within that specialty, however, patients could be represented in multiple specialties if treated across departments. The highest BMI registered in the study period was selected for the analyses. Patients with extremes in BMI (below 19 or exceeding 65 kg/m$^2$) were excluded from the analysis because of potential inaccuracies in registration. Additionally, patients from the MBS outpatient clinic were excluded, as they are managed through a dedicated care pathway not representative of general hospital populations.

### Data collection and classification

Relevant data were extracted from the hospital's electronic health record system, including BMI, DBC codes, and corresponding diagnoses. Patients were categorized into three BMI groups based on WHO classifications: normal weight (BMI 19–25 kg/m$^2$), overweight (BMI 25–35 kg/m$^2$), and severe obesity (BMI ≥ 35 kg/m$^2$). The analysis included all medical specialties except pediatrics, as the inclusion criteria were limited to adults. Specific focus was given to surgical, cardiopulmonary, internal medicine, and gastroenterology specialties, as these represented the largest clinical specialties in terms of patient volume and number of procedures in the hospital, and showed a high prevalence of severe obesity in preliminary analyses. To ensure interpretability and statistical reliability, prevalence figures for severe obesity were reported only when the rate within a DBC group exceeded 7%—twice the prevalence of obesity in the Dutch population (3.5%)—or when a clinical association between obesity and the diagnosis was evident, as based on existing literature or clinical consensus within the study team (e.g., diabetes mellitus, osteoarthritis, or sleep apnea). DBC groups with fewer than 10 patients were excluded to avoid unstable estimates.

### Procedure data extraction

To quantify healthcare utilization in patients with severe obesity, procedural data were extracted from the hospital's electronic health record system for all patients with severe obesity. Procedures were categorized by medical specialty, based

on standardized procedure codes linked to the DBC registrations. The total number of procedures per specialty was calculated for the 2022–2023 study period. Specialties with the highest procedural volumes in this patient group were selected for further analysis.

### Outcome measures

The primary outcome was the prevalence of severe obesity across hospital specialties, based on DRGs. A subgroup analysis was conducted in surgical, cardiopulmonary, internal medicine, and gastroenterology specialties. These groups were selected based on known associations between obesity and diagnostic or treatment complexity.

Secondary outcomes included the number and type of procedures performed in patients with severe obesity. Subgroup analyses were conducted within the four specialties with the highest volume of procedures in patients with severe obesity, e.g., cardiology, general surgery, cardiothoracic surgery, and orthopedic surgery. Furthermore, a subgroup analysis was performed to analyze the prevalence of severe obesity in patients undergoing hernia repair, given the high prevalence of obesity in this group.

### Statistical analysis

Descriptive statistics were used to summarize BMI distributions, reported as absolute numbers with percentages. Frequencies of severe obesity were calculated assuming a uniform distribution across specialties, using the observed average hospital prevalence of 8% as a reference. Differences in these reference counts of patients with severe obesity across specialties was evaluated using chi-square tests, with a significance threshold of $p < .05$. Subgroup analyses were performed to examine the frequency of specific procedures among patients with severe obesity. All analyses were conducted using Microsoft Excel 2021 (Microsoft Corp., Redmond, WA, USA).

### Ethical considerations

This study conformed to the ethical standards outlined in the 1964 Declaration of Helsinki and its subsequent amendments.[12] As it involved a retrospective analysis of anonymized patient data, formal ethical approval was not required under Dutch research regulations. Data confidentiality and patient privacy were strictly maintained throughout.

### Results

Among all medical specialties, patients with a normal weight accounted for 33.7% of all patients, while 58.3% of patients were classified as overweight, and 8.0% of patients met the criteria for severe obesity. The highest proportions of severely obese patients were found in pulmonary medicine (12.8%), internal medicine (10.4%), neurosurgery (10.3%), pain medicine (10.2%), and orthopedic surgery (9.8%). In these specialties, over 60% of patients had a BMI above $25 \, kg/m^2$. The complete BMI distribution per specialty is presented in Table 1. The prevalence of severe obesity was significantly higher than the hospital-wide average of 8.0% in specialties such as cardiology ($p < .001$), gastroenterology ($p < .001$), gynecology ($p < .001$), internal medicine ($p < .001$), neurosurgery ($p = .002$), ophthalmology ($p < .001$), orthopedic surgery ($p < .001$), pain medicine ($p < .001$), and pulmonary medicine ($p < .001$). Furthermore, the prevalence of severe obesity was significantly lower than the hospital-wide average of 8.0% in specialties such as cardiothoracic surgery ($p = .020$), geriatrics ($p < .001$), and radiotherapy ($p = .005$)

### Subgroup analyses of BMI distribution across surgical specialties

A focused subgroup analysis was performed on four surgical specialties— general surgery, urology, orthopedic surgery, and plastic surgery— due to the high volume of procedures and known technical or metabolic challenges associated with obesity in these fields. For each specialty, a selection of high-volume and obesity associated diagnoses was evaluated (Fig 1).

**Table 1. Number and percentage of patients treated at Catharina Hospital in 2022-2023 divided per BMI group and medical specialty.**

| Specialism | Total, n | BMI 19–25 kg/m², n (%) | BMI 25–35 kg/m², n (%) | BMI ≥ 35 kg/m², n (%) | Reference count of patients with BMI ≥ 35 kg/m², n* | p-value |
|---|---|---|---|---|---|---|
| *Cardiology* | 30,154 | 8,172 (27.1) | 19,310 (64.0) | 2,672 (8.9) | 2,412 | **<.001** |
| *Cardiothoracic surgery* | 8,325 | 2,111 (25.4) | 5,595 (67.2) | 619 (7.4) | 666 | **.020** |
| *Dermatology* | 9,115 | 3,102 (34.0) | 5,246 (57.6) | 767 (8.4) | 729 | .133 |
| *Gastroenterology* | 16,732 | 5,508 (32.9) | 9,763 (58.3) | 1,461 (8.7) | 1,339 | **<.001** |
| *General surgery* | 22,531 | 7,791 (34.6) | 12,884 (57.2) | 1,856 (8.2) | 1,802 | .090 |
| *Geriatrics* | 4,722 | 1,941 (41.1) | 2,505 (53.0) | 276 (5.8) | 378 | **<.001** |
| *Gynecology* | 13,431 | 5,514 (41.1) | 6,692 (49.8) | 1,225 (9.1) | 1,074 | **<.001** |
| *Internal medicine* | 21,525 | 6,707 (31.2) | 12,589 (58.5) | 2,229 (10.4) | 1,722 | **<.001** |
| *Neurology* | 21,205 | 7,078 (33.4) | 12,465 (58.8) | 1,662 (7.8) | 1,696 | .057 |
| *Neurosurgery* | 1,459 | 380 (26.0) | 929 (63.7) | 150 (10.3) | 117 | **.002** |
| *Ophthalmology* | 12,203 | 3,902 (32.0) | 7,127 (58.4) | 1,174 (9.6) | 976 | **<.001** |
| *Orthopedic surgery* | 13,873 | 4,183 (30.2) | 8,329 (60.0) | 1,361 (9.8) | 1,110 | **<.001** |
| *Otorhinolaryngology* | 9,130 | 3,024 (33.1) | 5,337 (58.5) | 769 (8.4) | 730 | .127 |
| *Pain medicine* | 4,968 | 1,429 (28.8) | 3,034 (61.1) | 505 (10.2) | 397 | **<.001** |
| *Plastic surgery* | 5,899 | 1,985 (33.6) | 3,427 (58.1) | 487 (8.3) | 472 | .862 |
| *Psychiatry* | 271 | 107 (39.5) | 140 (51.7) | 24 (8.9) | 22 | .618 |
| *Pulmonary medicine* | 12,813 | 3,448 (26.9) | 7,721 (60.3) | 1,644 (12.8) | 1,025 | **<.001** |
| *Radiotherapy* | 5,624 | 1,964 (34.9) | 3,261 (58.0) | 399 (7.1) | 450 | **.005** |
| *Rehabilitation medicine* | 1,019 | 308 (30.2) | 620 (60.8) | 91 (8.9) | 82 | .293 |
| *Urology* | 11,008 | 3,349 (30.4) | 6,783 (61.6) | 876 (8.0) | 881 | .267 |

* Reference count of patients with BMI ≥ 35 kg/m2 assuming a hospital-wide 8.0% prevalence.

BMI; body mass index.

In each of the selected surgical diagnosis groups, over 60% of patients had a BMI exceeding 25 kg/m².

For example, in general surgery, 17.5% of patients treated for diabetic foot and 13.2% of those with diverticulitis had severe obesity. In orthopedic surgery, 17.7% of patients with knee osteoarthritis were severely obese. Urology and plastic surgery also showed elevated BMI distributions in conditions such as prolapse, urinary tract stones, and trigger finger.

### Subgroup analyses of BMI distribution across cardiopulmonary specialties

Across cardiopulmonary specialties, 73% of individuals had a BMI exceeding 25 kg/m².

In the field of cardiology, elevated BMI was particularly prevalent among patients with heart failure, atrial fibrillation, and hypertension (Fig 2). Specifically, 13.0% of those treated for heart failure, 9.6% for atrial fibrillation, and 10.3% for hypertension had a BMI in the severe obesity category.

Within pulmonary medicine, severe obesity was especially prominent among patients with chronic bronchitis (54.9%), gastroesophageal-related asthma (31.3%), and sleep-related disorders (22.2%).

### Subgroup analyses of BMI distribution across internal medicine and gastroenterology specialties

In both general internal medicine and gastroenterology, BMI distributions showed that overweight and obesity were more common than normal weight among patients. Across these specialties, a considerable proportion of patients had a BMI above the normal range with 68% of individuals exceeding a BMI of 25 kg/m².

In general internal medicine, severe obesity was most common among patients with diabetes mellitus (17.3%), gonadal dysfunction (17.0%), Addison's disease (16.1%), hypothyroidism (13.7%), renal insufficiency (13.7%), and hypertension (13.1%) (see Fig 3).

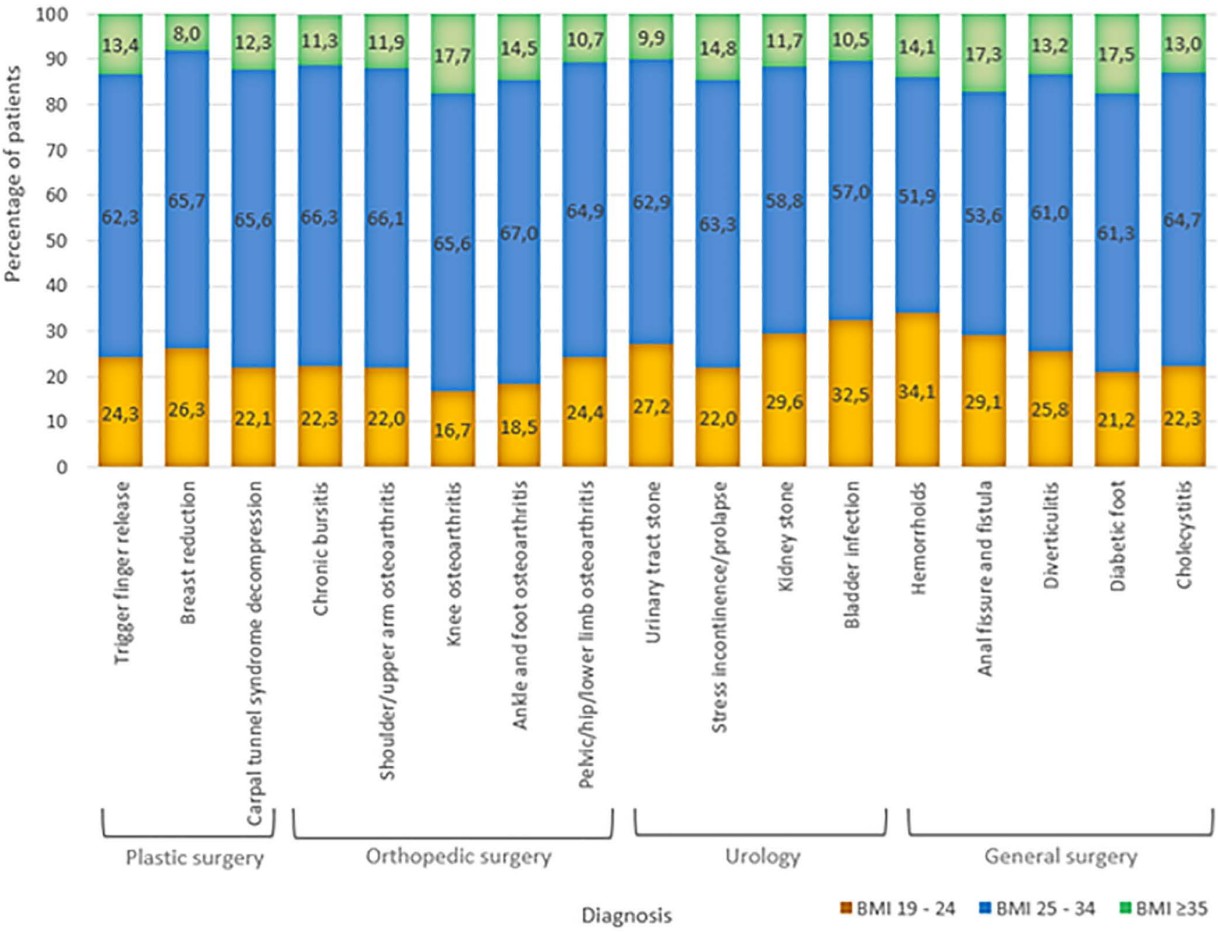

**Fig 1. BMI distribution across selected diagnoses in surgical specialties*.** * Selected surgical specialties include plastic surgery, orthopedic surgery, urology, and general surgery.

In gastroenterology, the prevalence of severe obesity varied across diagnoses. The highest proportions were observed in patients with benign liver tumors (17.3%) and liver cirrhosis (17.1%), followed by acute pancreatitis (14.5%), cholelithiasis (12.4%), hepatitis (12.1%), and diverticulitis (11.8%). Slightly lower rates were seen in patients diagnosed with ileus (11.0%), gastrointestinal bleeding (11.0%), abdominal pain (10.9%), and achalasia (10.8%).

## Prevalence of severe obesity in procedures and surgeries

A total of 5,472 surgical procedures were performed on patients with severe obesity in the two study years, this represents approximately 13.5% of all 40,475 procedures performed in these years in patients with a known BMI. The distribution of these procedures across medical specialties is summarized in Table 2. Cardiology, general surgery, and cardiothoracic surgery accounted for the highest procedure volumes, with 1,387, 1,121, and 756 procedures respectively. Additionally, orthopedic surgery and gynecology represented substantial shares with 743 and 359 procedures respectively.

Additionally, hernia-related diagnoses were analyzed separately. Incisional and umbilical/epigastric hernias showed a strong association with severe obesity (16.1% and 12.4%, respectively), while inguinal and femoral hernias were more prevalent in patients with lower BMI (Table 3).

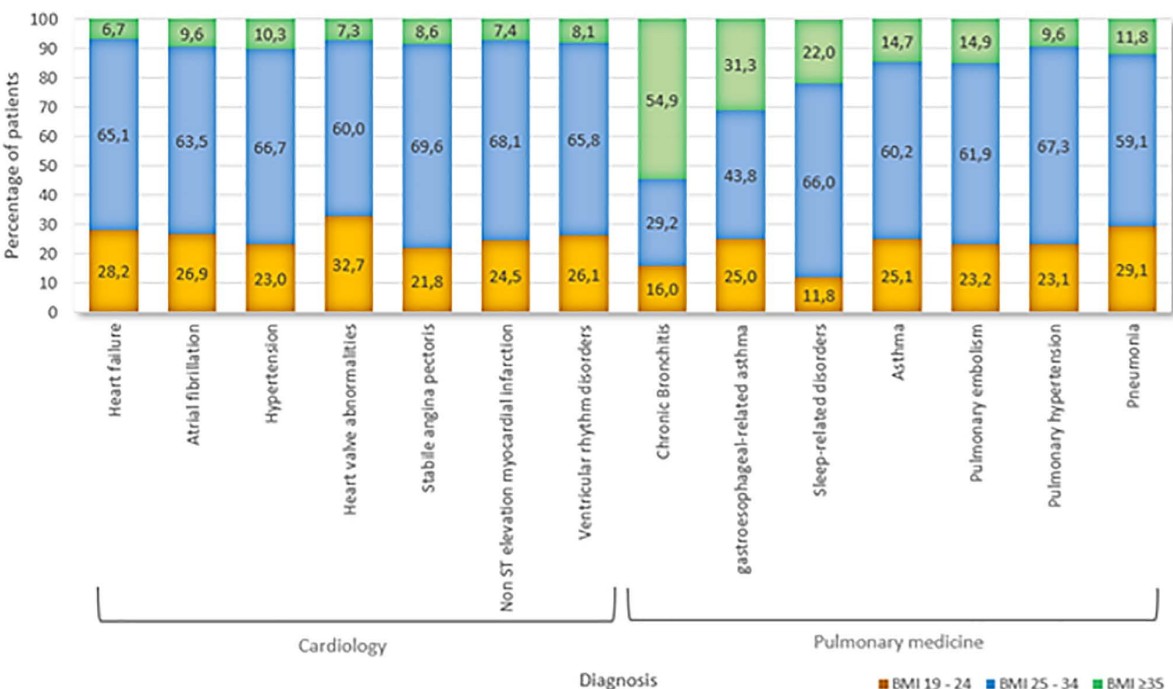

**Fig 2. BMI distribution across cardiopulmonary diagnoses.**

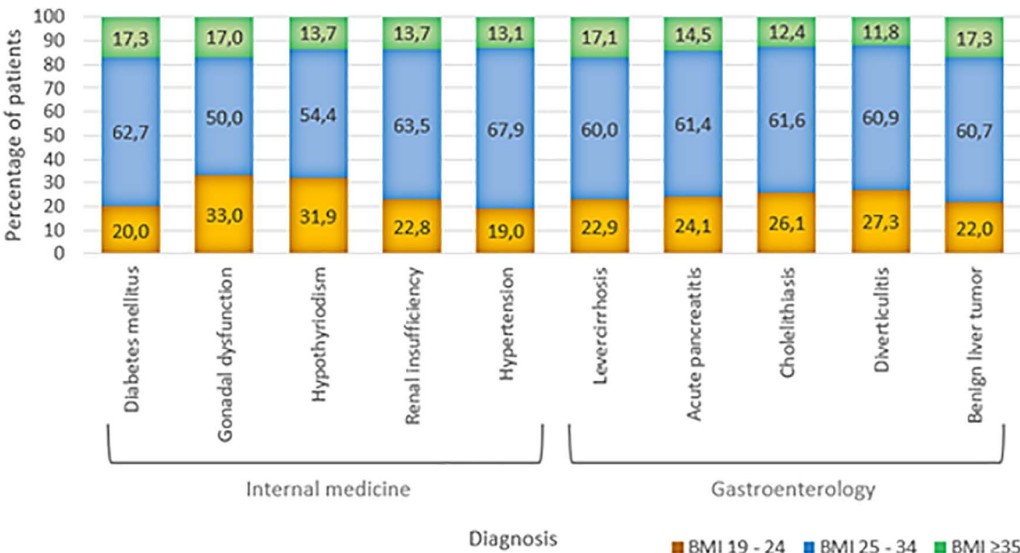

**Fig 3. BMI distribution across internal medicine and gastroenterology diagnoses.**

## Discussion

This study demonstrates that severe obesity is disproportionately represented within hospital populations, with a prevalence of 8.0%, compared to the national average of 3.5% in the general Dutch population.[4] The prevalence of severe

**Table 2. Number of procedures per specialism in patients with severe obesity.**

| Specialism | Number of procedures, n |
| --- | --- |
| Cardiology | 1,387 |
| General surgery | 1,121 |
| Cardiothoracic surgery | 756 |
| Orthopedic surgery | 743 |
| Gynecology | 359 |
| Urology | 349 |
| Ophthalmology | 322 |
| Plastic surgery | 209 |
| Otorhinolaryngology | 98 |
| Neurosurgery | 56 |
| Pain medicine | 53 |
| Oral and maxillofacial surgery | 19 |

The procedures in cardiology (n = 1,387) primarily involved percutaneous coronary interventions and angiographies (Fig 4). General surgery (n = 1,121) included a wide range of abdominal procedures, with laparoscopic cholecystectomy and appendectomy being the most common. Cardiothoracic surgery (n = 756) was dominated by CABG and valve replacements, reflecting a standardized procedural portfolio. In orthopedic surgery (n = 743), the majority of procedures addressed degenerative joint disease, with knee and hip replacements accounting for most cases.

obesity was particularly high in pulmonary medicine, internal medicine, and surgical specialties. These findings suggest that patients with severe obesity utilize hospital care at disproportionately high rates, hypothetically due to increased multimorbidity, procedural complexity, and disease severity.

Previous studies have shown strong associations between obesity and adverse health outcomes, including prolonged hospital stays and postoperative complications [2,13,14]. Our results extend this evidence by quantifying the distribution of severe obesity across hospital specialties and identifying diagnoses and procedures with the highest obesity prevalence. The highest rates were found in pulmonary medicine, particularly among patients with chronic bronchitis, gastroesophageal-related asthma, and sleep-related disorders. These findings are consistent with prior literature linking obesity to obstructive sleep apnea, obesity hypoventilation syndrome, and obstructive lung disease [7,8,15]. The bidirectional relationship between sleep fragmentation and metabolic dysregulation further reinforces the need for integrated pulmonary and metabolic care. Within cardiovascular specialties, high rates of obesity were observed among patients undergoing coronary artery bypass grafting, valve replacement, and surgical ablation. These findings support existing evidence that excess adiposity contributes to myocardial remodeling, epicardial fat accumulation, systemic inflammation, and subsequent cardiac dysfunction [16–18]. Goh et al. similarly reported a high prevalence of obesity among cardiothoracic surgery patients, along with increased risks for complications and poor perioperative outcomes [19]. Furthermore, Balayah et al. demonstrated that central adiposity markers were associated with increased in-hospital and one-year mortality in patients with acute coronary syndrome, highlighting the prognostic relevance of obesity in cardiac populations [20].

Obesity also significantly impacted surgical specialties. A high prevalence of severe obesity was noted among patients undergoing incisional and umbilical/epigastric hernia repair, consistent with earlier research suggesting that increased intra-abdominal pressure and impaired fascial integrity elevate hernia risk [21,22]. Similarly, in orthopedic surgery, severe obesity was common among patients with knee and hip osteoarthritis. These findings align with literature showing that excess weight accelerates joint degeneration and increases perioperative risks, particularly in joint replacement procedures [23–25].

**Table 3. Sub analysis of hernia-related diagnosis in patients with severe obesity.**

| Diagnosis | BMI ≥ 35 kg/m², n (%) |
|---|---|
| Hernia cicatricialis | 58 (16.1) |
| Hernia umbilicalis/epigastrica | 48 (12.4) |
| Hernia diaphragmatica | 18 (5.0) |
| Hernia femoralis/inguinalis | 27 (3.5) |

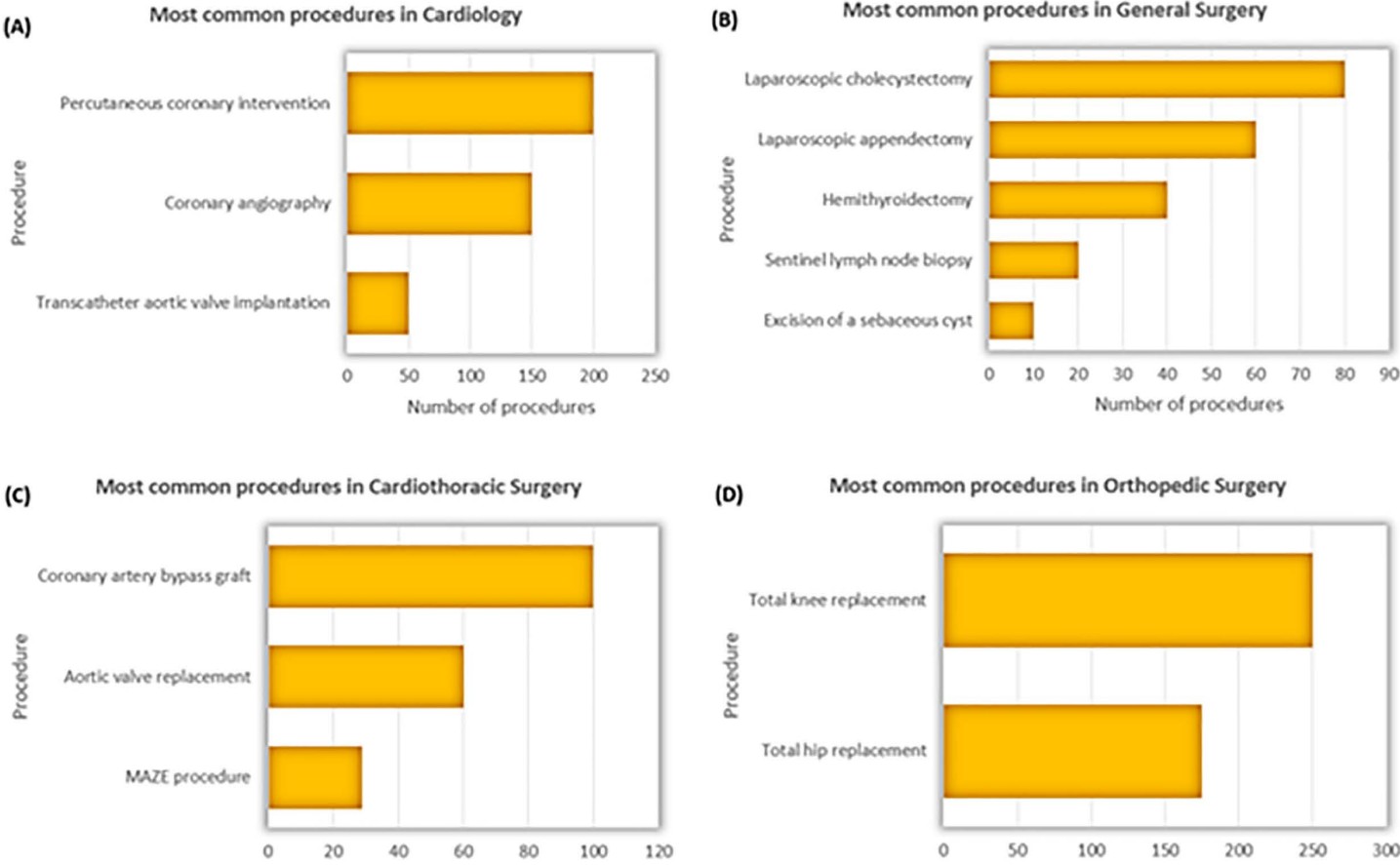

**Fig 4. Most common surgical procedures in cardiology (A), general surgery (B), cardiothoracic surgery (C), and orthopedic surgery (D) in patients with severe obesity.**

Despite the consistency of this study's results with prior literature, there are several limitations that should be acknowledged. This was a retrospective, single-center study, limiting causal inference and generalizability to other hospital types. Definitions of comorbidities were based on registry coding without biochemical verification, which may have led to under-reporting. Selection bias could have occurred, as only patients with complete BMI data and active DBC registrations were included. Additionally, we excluded patients managed at the MBS outpatient clinic and had no access to post-admission measures of central adiposity, such as waist circumference or body composition, which could have provided a more accurate stratification.

Furthermore, while our analysis is based on BMI as a proxy for adiposity, the Lancet Commission on the Definition and Diagnosis of Clinical Obesity has emphasized that clinical obesity should be defined by the presence of excess or dysfunctional adiposity that impairs health, irrespective of BMI.[1] This more nuanced approach may better capture the health risks associated with obesity and suggests that BMI-based prevalence figures, including those reported in our study, may underestimate the true clinical burden of obesity in hospital settings. Lastly, age was not included as a potential confounder, although the prevalence of obesity in the Netherlands is known to increase with age up to approximately 65 years [26].

Nonetheless, this study has several strengths. This is one of the first studies exploring the population of patients living with severe obesity in a hospital setting. The use of a large, real-world hospital dataset enables broad applicability to tertiary care settings. The exclusion of MBS and revisional cases helped define a more homogeneous cohort reflective of general specialty care. Structured DBC coding allowed for precise mapping of obesity prevalence across departments and procedures.

These findings underscore the substantial presence of patients with severe obesity in non-bariatric hospital care, particularly in cardiopulmonary, surgical, and internal medicine departments. By highlighting the disproportionate procedural demand in this group, this study adds to the growing awareness that obesity is a hospital-wide issue — not limited to bariatric clinics. This may inform future research, resource allocation, and the development of tailored clinical pathways.

Future research should further explore the clinical outcomes, resource utilization, and complication rates in patients with severe obesity across specialties. Particular attention is needed for specialties disproportionately affected – such as cardiopulmonary, surgical, and orthopedic specialties – to assess how obesity-specific challenges manifest in different clinical contexts. Longitudinal studies could help clarify causal relationships between obesity and hospital-based morbidity, and evaluate the effectiveness of targeted interventions, such as weight management programs or tailored perioperative protocols. This knowledge could inform the development of integrated, specialty-specific obesity strategies that incorporate early BMI screening, perioperative optimization, and multidisciplinary chronic disease management.

In conclusion, severe obesity is markedly overrepresented across hospital specialties, with the greatest impact observed in cardiopulmonary, surgical, and orthopedic fields. Addressing the burden of obesity within hospital systems is crucial for improving outcomes, enhancing patient safety, and ensuring sustainable healthcare delivery.

## Supporting information

**S1 File. Operations per specialism.**
(XLSX)

**S2 File. Analysis.**
(XLSX)

**S3 File. Patients per specialisme 2022–2023.**
(XLSX)

## Author contributions

**Conceptualization:** Ephrahim E. Jerry, Miriam A. Scheurwater, Janine J.P. Ghielen, Simon W. Nienhuijs.

**Formal analysis:** Ephrahim E. Jerry, Miriam A. Scheurwater.

**Supervision:** Simon W. Nienhuijs.

**Writing – original draft:** Ephrahim E. Jerry, Miriam A. Scheurwater.

**Writing – review & editing:** Janine J.P. Ghielen, Ilse van Donkelaar, Dennis van Veghel, Lukas R.C. Dekker, Simon W. Nienhuijs.

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
