## [Decision Letter · Decision Letter 0]

3 Aug 2025

PONE-D-25-34933Exploring the prevalence of severe obesity in a large teaching hospitalPLOS ONE

Dear Dr. Jerry,

Thank you for submitting your manuscript to PLOS ONE. After careful consideration, we feel that it has merit but does not fully meet PLOS ONE’s publication criteria as it currently stands. Therefore, we invite you to submit a revised version of the manuscript that addresses the points raised during the review process.

**This is a very interesting and methodologically sound research. Please address all reviewers’ comments before proceeding to final evaluation.**==============================

We look forward to receiving your revised manuscript.

Kind regards,

Athanasios G. Pantelis

Academic Editor

PLOS ONE

2. In the online submission form, you indicated that [The data underlying the results presented in the study are available from the corresponding author].

Additional Editor Comments (if provided):

Reviewers' comments:

Reviewer's Responses to Questions

**Comments to the Author**

1. Is the manuscript technically sound, and do the data support the conclusions?

Reviewer #1: Yes

Reviewer #2: Yes

2. Has the statistical analysis been performed appropriately and rigorously? 

Reviewer #1: Yes

Reviewer #2: Yes

3. Have the authors made all data underlying the findings in their manuscript fully available?

Reviewer #1: No

Reviewer #2: Yes

4. Is the manuscript presented in an intelligible fashion and written in standard English?

Reviewer #1: Yes

Reviewer #2: Yes

5. Review Comments to the Author

Reviewer #1: Dear Authors,

I have read your manuscript and found it impressive. I do, however, have a few questions and suggestions that I hope could enhance the quality and impact of your upcoming manuscript:

1- The Lancet Commission on the Definition and Diagnosis of Clinical Obesity recommends defining and diagnosing clinical obesity based on the presence of "excess or dysfunctional adiposity that impairs health," irrespective of body mass index. I suggest incorporating this definition into both the Introduction and Discussion sections of your manuscript, as it may influence the accurate estimation of obesity prevalence in your study population.

2- An important subgroup worth considering is patients with cancer, particularly those with obesity-related cancers. Including this subgroup in your analysis could provide valuable insights and strengthen your findings. Would it be possible to add this population to your study cohort

Reviewer #2: Changes according to IFSO's accepted terminology

1. "Bariatric Surgery" → "Metabolic Bariatric Surgery (MBS)"

• Line 36

Original: Patients with a history or current treatment for bariatric surgery were excluded.

Suggested: Patients with a history or current treatment for Metabolic Bariatric Surgery (MBS) were excluded.

• Line 120

Original: Additionally, patients from the bariatric outpatient clinic were excluded

Suggested: Additionally, patients from the Metabolic Bariatric Surgery (MBS) outpatient clinic were excluded

• Line 321

Original: we excluded patients managed at the bariatric outpatient clinic

Suggested: we excluded patients managed at the Metabolic Bariatric Surgery (MBS) outpatient clinic

• Line 327

Original: The exclusion of bariatric and revisional cases helped define

Suggested: The exclusion of Metabolic Bariatric Surgery (MBS) and revisional cases helped define

• Line 415 (Reference)

Original: adult and adolescent bariatric surgery candidates.

Suggested: adult and adolescent Metabolic Bariatric Surgery (MBS) candidates.

2. "Comorbidities" → "Obesity-associated diseases/disorders"

• Line 314

Original: Despite the consistency of this study's results with prior literature, there are several limitations that should be acknowledged.

(No change needed here — included for context)

• Line 321

Original: we had no access to post-admission measures of central adiposity, such as waist circumference or body composition, which could have provided a more accurate stratification.

(Also acceptable — contextually fine)

• Line 415

Original: The prevalence of co-morbidities in adult and adolescent bariatric surgery candidates.

Suggested: The prevalence of obesity-associated diseases in adult and adolescent Metabolic Bariatric Surgery (MBS) candidates.

Optional: Add IFSO-aligned BMI Classification

While “severe obesity” (BMI ≥35 kg/m²) is used correctly, IFSO recommends using specific classes:

• BMI 35–<40: Obesity II

• BMI 40–<50: Obesity III

• BMI 50–<60: Obesity IV

• BMI ≥60: Obesity V

Overall , provides important data concerning BMI and hospital admittance and can be the start of further research

6. PLOS authors have the option to publish the peer review history of their article (what does this mean? ). If published, this will include your full peer review and any attached files.

**Do you want your identity to be public for this peer review?** For information about this choice, including consent withdrawal, please see our Privacy Policy .

Reviewer #1: **Yes: ** Mohammad Kermansaravi

Reviewer #2: No

---

## [Author Response · Author response to Decision Letter 1]

1 Sep 2025

Dear Editors,

In response to your request, The underlying data have been provided as a supplementary file during resubmission in accordance with the PLOS ONE data availability policy.

Kind regards,

Ephrahim Jerry

---

## [Editor Report · Decision Letter 1]

3 Sep 2025

Exploring the prevalence of severe obesity in a large teaching hospital

PONE-D-25-34933R1

Dear Dr. Jerry,

We’re pleased to inform you that your manuscript has been judged scientifically suitable for publication and will be formally accepted for publication once it meets all outstanding technical requirements.

Kind regards,

Athanasios G. Pantelis

Academic Editor

PLOS ONE
---

## [Editor Report · Acceptance letter]

PONE-D-25-34933R1

PLOS ONE

Dear Dr. Jerry,

I'm pleased to inform you that your manuscript has been deemed suitable for publication in PLOS ONE. Congratulations! Your manuscript is now being handed over to our production team.

Kind regards,

on behalf of

Dr. Athanasios G. Pantelis

Academic Editor

PLOS ONE